# New Evidence for the Bronze Age Zooarchaeology in the Inland Area of the Iberian Peninsula through the Analysis of Pista de Motos (Villaverde Bajo, Madrid)

**DOI:** 10.3390/ani14030413

**Published:** 2024-01-26

**Authors:** Verónica Estaca-Gómez, Rocío Cruz-Alcázar, Silvia Tardaguila-Giacomozzi, José Yravedra

**Affiliations:** 1Departamento de Prehistoria, Historia Antigua y Arqueología, Facultad de Geografía e Historia, Universidad Complutense (UCM), C/Profesor Aranguren, 28040 Madrid, Spain; vestaca@ucm.es (V.E.-G.); roccruz@ucm.es (R.C.-A.); 2Research Group: Arqueología Prehistórica, Universidad Complutense (UCM), C/Profesor Aranguren, 28040 Madrid, Spain; 3Wolfson College, University of Oxford, Linton Rd., Oxford OX2 6UD, UK; silviatagia@gmail.com; 4Unidad del CAI de Arqueometría y Análisis Arqueológico, Universidad Complutense (UCM), C/Profesor Aranguren, 28040 Madrid, Spain; 5Research Group: Ecosistemas Cuaternarios, Facultad de Ciencias Geológicas, Universidad Complutense (UCM), 28040 Madrid, Spain

**Keywords:** zooarchaeology, bronze age, middle taggus, Iberian Peninsula, faunal deposits

## Abstract

**Simple Summary:**

In this paper, we present the zooarchaeological and taphonomic study of a new Bronze Age site in the inland of the Iberian Peninsula. This study represents a significant contribution within the studied area, as it analyses the Pista de Motos faunal collection, one of the few representative samples from this period. This paper significantly contributes to completing the scarce information available for the Bronze Archaeology in the Middle Tagus Valley, located inland of the Iberian Peninsula.

**Abstract:**

The Bronze Age zooarchaeological research for the interior and other regions of the Iberian Peninsula is currently limited. Despite several sites with known zooarchaeological profiles from the period, the main issue is that most of these derive from fragmentary and unrepresentative faunal records or are biased profiles from old excavations. New work has yielded novel zooarchaeological results in recent years that could help fill the existing zooarchaeological information gap in the Iberian inland, particularly in the Middle Tagus Valley. However, these projects are in the initial research stages and have not published much of their results. This paper presents the zooarchaeological profile of the Pista de Motos Bronze Age site to help fill this information gap. It analyses the taxonomic representation, skeletal profiles, and human activity patterns associated with faunal use. These observations suggest that animal exploitation at the site followed two primary purposes. One was linked to economic activities, mainly to obtain meat, milk, wool, or animal labour. The other was probably associated with symbolic-ritual practices suggested by the complete animal burials in some excavated units. We contextualise these interpretations with evidence from other Bronze Age sites in the Middle Tagus Valley. Finally, the paper assesses to what extent Pista de Motos is a relevant site for the zooarchaeology of the Bronze Age in the Iberian inland.

## 1. Introduction

The inland of the Iberian Peninsula, specifically the Middle Tagus Valley (the Madrid–Guadalajara and Toledo area), is a region known since the 19th century for its considerable archaeological record [1]. Despite the abundant regional archaeological sites representing all historical periods, published zooarchaeological profiles are scarce. This limited zooarchaeological research has already been addressed for the regional Neolithic [2,3], Chalcolithic [4,5], and Iron Age [6] but not so much for the Bronze Age period.

In recent years, there have been considerable efforts to overcome these limitations, either with novel research into prehistoric Age sites [7,8,9] or by excavating new settlements such as Camino de las Yeseras [10,11,12,13,14,15,16,17], Val de la Viña [18,19], Las Zanjillas [20], 718-05-H-04, also known as Pista de Motos [21], or Humanejos [22,23]. Although these further the regional Chalcolithic and Bronze Age zooarchaeological profiling, our knowledge remains limited. Research on sites such as Aldovea, Barrio del Castillo, Humanejos, or la Cuesta have yet to be published, and others such as Camino de las Yeseras are still being researched and have not yielded complete results. For this reason, none of the currently published Bronze Age zooarchaeological records exceeds 1000 faunal remains. Fábrica de Ladrillos is currently the only site with 2949 remains, including determinable and indeterminate fauna [24].

Due to the work carried out in recent years, there is an increasing amount of zooarchaeological profiles available for the Iron Age [7,8,9] and Chalcolithic period in the Middle Tagus Valley. Still, most of these sites lack materials from the Bronze Age. Research at Barranco del Herrero [25], Soto del Henares [26], Las Cabeceras [27], Aguas Vivas [28,29], Entreviñas I [30], Las Zanjillas [31], or Aldovea [32] attests to this. Older studies from sites like Angosta de los Mancebos [33], Esgaravita [34], Fábrica de Euskalduna, Capricho, and Loma Chiclana [4], Juan Barbero [35], Huerta de los Cabreros [36], Espinillo [37], El Ventorro [38,39,40], and Las Matillas [41] complete the available data to understand the zooarchaeology of the Middle Tagus valley. This zooarchaeological analysis of Pista de Motos deposits contributes to enlarging the knowledge of agropastoral strategies during the Bronze Age in the Middle Tagus Valley.

This critical situation highlights the dire need to enlarge the Bronze Age zooarchaeological studies in the interior of the Middle Tagus Valley. Other Iberian Peninsula regions, such as the northwest, including Cantabria and the south of Portugal, also show a significant need for zooarchaeological samples from this period. Only analysing new and representative zooarchaeological sets will lead to a comprehensive understanding of the evolution of Iberian agricultural societies, from the Chalcolithic to the Romanization.

## 2. Pista de Motos and a Regional Overview

The Iberian Peninsula encompasses Spain and Portugal, which are further classified into various distinctive regions. The primary ones are the Cantabrian coast in the North, the Mediterranean region stretching from Catalonia to Andalusia in the East, the Atlantic area covering Portugal in the West, and the interior territories comprising the Duero, Tagus, and Guadiana valleys (Figure 1).

Site 718-05-H-04, also known as Pista de Motos, is located on the right bank of the Manzanares River, approximately 574 m above sea level, in the Villaverde municipality [21] (Figure 1). Pista de Motos includes several occupational phases from the Neolithic to the Middle Ages, with the Bronze Age being the most prominent. This period can be classified further into two distinct phases. The first is characterised by Bell Beaker pottery, with 2717 sherds found in 16 silos or pits. The second, associated with the Middle Bronze Age, has been identified in 192 silos or pits containing a total of 23,669 protocogotas ceramic sherds.

Domínguez-Alonso and Virseda [21] have provided only five radiocarbon dates obtained at the CIRCE laboratory in Naples. Despite the limited chronometric information available, we propose a simple two-phased Bayesian age model (Figure 2) to estimate both phases of the Bronze Age’s start and end boundaries (Table 1). The Bell Beaker period, defined by two radiocarbon dates (SU1251 and SU2417), yields a start boundary of 4546–4011 cal BP and an end boundary of 4219–3691 cal BP. Three radiocarbon dates define the Middle Bronze Age. SU962 corresponds to a multiple burial. SU2581 is associated with Protocogotas ceramics. Finally, SU1660 includes successive deposits of a pig, a dog, a goat, and a raven in anatomical connection. Thus, the Middle Bronze Age modelled start boundary is 3901–3407 cal BP, and the modelled end boundary is 3384–2933 cal BP. 

This preliminary chronology (Figure 2) should be further refined with additional radiocarbon dates, which will help constrain the phases further. Additional information about the original dates would also aid in evaluating the quality of the chronology. For instance, the dated material (in the case of bones, the C: N ratios should also be reported) and associations with the fauna should be clarified.

Pista de Motos is a typical illustration of the so-called “Campo de Hoyos” of the Middle Tagus Valley [5,42,43]. These are characterised by several negative features, such as pits and silos, which would have been used to store grain and later repurposed for garbage disposal. The presence of these silos and lithic industry, such as mills or sickle teeth, suggests that the site had an agro-pastoral character specialising in cereal exploitation [21]. Additional artefacts such as bone awls, found among the bone industry, or loom weights and cheese-making vessels, found among the ceramics, suggest additional economic practices to obtain wool and milk.
Figure 2Previously published radiocarbon dates [21] in a sequential two-phased Bayesian age model. A General Outlier model [44] was used because the dated material is unknown, modelled using OxCal v.4.4.4. [45] and calibrated against IntCal20 [46]. No laboratory codes were provided in the original publication [21].
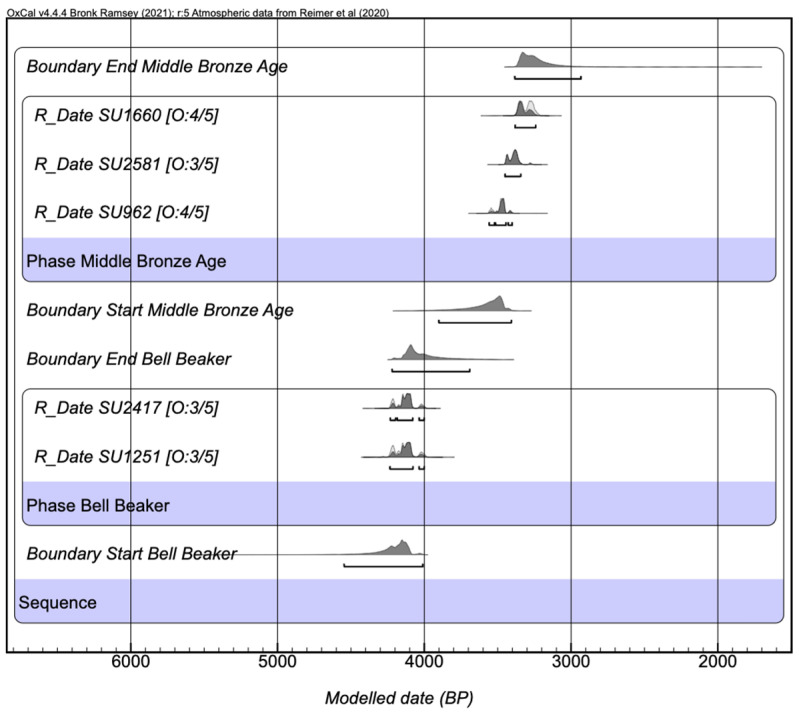



### 2.1. Materials

During the 2006 archaeological excavations, Pista de Motos yielded 8766 bone fragments, all discovered in silos and pits (Table 2). While the Bell Beaker Bronze phase is represented by 959 bone fragments, the Middle Bronze Age phase yielded 6859 remains. Thus, the latter is the most significant period for the site (Table 2). Due to this reason, it is also the main focus of this research. This study has identified from the osteological record the following species: *Ovis aries*, *Capra hircus*, *Canis familiaris*, *Sus domesticus*, *Equus ferus*, *Bos taurus*, *Sus scrofa*, *Cervus elaphus*, *Capra pyrenaica*, *Lepus* sp., and *Oryctolagus cuniculus*. The remains of various bird species, such as *Tyto alba*, *Alectoris rufa*, and *Corvus* sp., and amphibians, like *Bufo bufo*, were also present. Nonetheless, this paper primarily analyses macrovertebrates due to their predominant representation.

It is worth emphasising that although this study focuses on the Bronze Age, material culture from other periods (i.e., Neolithic, Iron Age, and Medieval) has also been found at the site (See Table 2 and Table 3). While this paper does not analyse the scarce zooarchaeological material from these periods, future papers focusing on these chronologies should assess them.

### 2.2. Methods: Zooarchaeology and Taphonomy

The identification of species featured in this study has taken as reference the faunal collections from the Department of Prehistory, Ancient History, and Archeology at the Complutense University of Madrid. We also referred to atlases such as those by [47,48]. For a more specific identification of species, like sheep and goats, we followed [49,50,51,52]. To particularly distinguish between *Capra hircus* and *Capra pyrenaica*, we followed [52], and for the differentiation between pig and wild boar, we observed [53].

We have identified the taxonomically undetermined but anatomically identifiable bones following the methodology outlined by Yravedra (2006) [54]. Pista de Motos’ remains have been categorised as large animals (*Bos taurus*, *Equus ferus*), medium-sized animals (*Cervus elaphus*, or juvenile *Bos/Equus*), or small animals (*Ovis/Capra*, *Sus domesticus*). In instances where bones could not be taxonomically assigned or categorised by size, they have been considered indeterminate.

We have identified the remains using the NISP (Number of Identified Specimens taxonomically or anatomically) and the MNI (Minimum Number of Individuals). Based on the most abundant anatomical remains, these have been assigned, considering laterality, age, sex, and size following Brain’s (1969) [55] methodology.

Age mortality patterns have been defined based on dental emergence and dental wear for goats [56], sheep [57,58], cows [59,60], and pigs [61]. Age estimation from epiphyseal fusion follows the recommendations of Barone (1999) [62]. Following the authors mentioned above for goats, pigs, sheep, and cows, we can estimate mortality patterns in months, which allows us to make approximate relative seasonal estimates.

Although all these methods present some limitations, as they can be subject to variability, the reality is that most authors follow them. Considering this, the conclusions of these studies are reliable.

Skeletal profiles have been defined following the divisions in [54] for cranial, appendicular, and axial elements. Taphonomic analysis has involved studying the bone surfaces using 20X hand lenses following [63,64]. After evaluating their condition, we examined the bone weathering further, following the guidelines of [65]. We have also assessed the impact of biological alterations, highlighting processes caused by carnivores such as pits, scores, and punctures, which we have identified as tooth marks according to [64,66,67,68]. We have also examined anthropogenic alterations, identifying cut marks according to [66,68,69]. Despite not being reported at the site to identify percussion marks, we have followed [64,67]. Lastly, we have documented the degree of burnt bones using the categories described by [70]. Beyond the analysis of bone surfaces, we have considered bone fracturing, distinguishing between fresh and dry fractures following the criteria of [71].

## 3. Results: Zooarchaeological and Taphonomic

### 3.1. Taxonomical Representation and Mortality Profiles

The bone assemblage at Pista de Motos, totalling 8766 remains dating from various chronological periods, offers valuable insights. Although this study focuses on the Bronze Age, it is noteworthy to mention the material from the other periods briefly. Only 36 remains from the Neolithic were found, predominantly consisting of domestic species, primarily characterised by mixed herds of sheep and goats. There were also identifiable but insignificant samples from the Iron Age, Visigoth, and Islamic periods. Caprines were the most prominent group in all of them. Additionally, a prehistoric fauna set could not be assigned to any specific period (Table 2 and Table 3).

Moving to the Bronze Age, starting with the Bell Beaker phase, there is a sample of 959 remains, where domestic fauna prevails over wild fauna. *Bos taurus* was the most abundant species, followed by caprines. Other domestic animals, such as pigs and dogs, were also present along with wild species, including deer, horses, wild boars, and rabbits (Table 2).

The Middle Bronze Age is characterised by the intentional deposition of animals in anatomical connection found in silos across different Stratigraphic Units (SU). Respectively, SU 1361 contained a barn owl, SU 1666 a raven, SU 1662 a pig, SUs 3171 and 981 a cow, SUs 1661 and 1663 a dog, SUs 1664, 2692, and 2693 a domestic goat, and SU 857 a wild Iberian ibex on a votive deposit.

Considering the MNI, cows are the most prominent taxon during the Bell Beaker phase, followed by caprines represented by sheep and goats, and finally, pigs. There is less representation of other animals like dogs or horses (Table 3). During the Middle Bronze Age, mixed herds of sheep and goats were the most abundant taxa, followed by cows, pigs, and dogs. Among caprines, goats are more prominent than sheep (Table 3).

According to observations from the NISP and MNI in Table 2 and Table 3, Pista de Motos had an economy majorly centred around livestock. This suggests that the site’s inhabitants mainly exploited herds of caprines, bovines, and pigs. The relevance of other animals, such as dogs, will be discussed later. Biometric analyses conducted for this animal have allowed us to differentiate between three types according to size. Two medium and one small-sized dogs were found (Appendix A).

Wild fauna, including horses, deer, wild Iberian ibex, wild boars, rabbits, and some birds, is scarcely represented, only limited to one individual each for horses, deer, the birds, and wild Iberian ibex. Two individuals were found for wild boar and four for rabbit (Table 3). The barn owl, the raven (Units SU 1666 and SU 1361), and the young wild Iberian ibex (Unit SU 857) were complete and buried in anatomical connection.

A mortality patterns analysis for the Bronze Age at Pista de Motos reveals differences between the Bell Beaker and Middle Bronze Age periods (Figure 3). In the Bell Beaker period, immature individuals, represented by infants and juveniles, were abundant. Cows represented 50%, pigs 60%, and goats 33% (Table 3 and Figure 3). In contrast, the Middle Bronze Age presented a predominance of adult individuals in all species except pigs, where juveniles and infants remained more prominent (Figure 3). Wild animals were only represented by adult individuals in both periods, except for the Iberian ibex, which was a juvenile (Table 3).

We have also estimated a preliminary time of death based on dental emergence and dental wear. Future studies could refine our current observations by including methods like stable isotope analysis, cement chronology, or dental microwear. Cows displayed two different mortality patterns according to the two chronological periods. While during the Bell Beaker phase, the mortality predominantly occurred between spring and summer, during the Middle Bronze Age, the mortality had a higher continuity with a preference towards the period between summer and autumn (Figure 3). A stable mortality is also observed in pigs, being more frequent between summer and autumn in both periods. Caprines’ mortality was generally concentrated in autumn, yet sheep and goats presented slightly different patterns. While the former had a higher mortality in autumn, the latter displayed a more continuous pattern, similar to that of pigs (Figure 4).

When analysing the taxonomic representation from each unit, it is observed that, despite some exceptions, most of the SUs contain only a few unrepresentative remains for each found species (refer to Appendix A). In contrast, SUs 1361, 1666, 1661, 1662, 1663, 1664, 2692, 2693, 3171, 857, 981, and 2371 present remains of an individual either entirely or partially in anatomical connection (Table 4). Some SUs stand out for having more than 100 NISPs of various species per SU (Table 4). Lastly, three SUs display remains associated with human burials. SUs 962 and 963 present a cow’s remains, and SU 1401 presents a goat’s remains, all accompanied by a few undetermined remains. However, these faunal–human associations are considered to be arbitrary, as faunal remains are incomplete and lack intentional depositions, which would have been found if faunal remains were in articulation.

Most animals deposited in anatomical connection are complete individuals (refer to the first column of Table 4). Among those incomplete, there are SU 1691, with a dog’s skull, and SU 981, which includes a cow’s skull, found alongside the wild Iberian ibex. Finally, there is a goat in SU 2693 and a cow in SU 3171 and SU573, respectively, represented by appendicular and/or axial elements in anatomical connection.

Although all the SUs with relevant faunal deposits generally present individuals of all ages, there are some exceptional cases. For instance, dogs are only represented by adult individuals (SU 1018, SU 1661, SU 1663). There is also a neonate pig in SU 2371 and an infant one in SU 1662. Several infant goats were found in SUs 1694 and 2693. Finally, two cow individuals were under 20 months old in SU 3171, and a juvenile was found in SU 573, alongside the remains of what could be an ox.

Following these observations, the generic taxa quantification described in Table 3, classified by SUs based on the MNI, could be refined further (see Table 4 and Table 5). Table 5 introduces a second quantification methodology based on the results of independently quantifying the MNI from each unit. According to this latter method, the cow emerges as the most representative animal, having a higher MNI than caprines, due to cows appearing in more SUs, thus increasing their final representativeness (Table 5).

When analysing the data from the SUs in Appendix A, it becomes apparent that only a few SUs have more than one individual per unit. Only six SUs have at least six individuals, with SU 3021 standing out as exceptional with ten individuals from different species. The SUs with a higher MNI are SU 1251, SU 1336, SU 2101, SU 2236, SU 2391, and SU 3021, with cows being the most abundant. SU 1251 is an exception, presenting goats and rabbits as the most prominent taxa. Consequently, cattle and caprines were the main animals during the Bronze Age at Pista de Motos, followed by pigs, dogs, and other wild species. Due to the limited representativeness of most SUs, these taxonomic assessments should be interpreted globally.

### 3.2. Analysis of the Skeletal Profiles

Regarding the skeletal profiles, complete animals are only found in a few SUs. Complete birds were discovered in SUs 1666 and 1361, goats in SUs 851, 1664, and 2692, pigs in SU 1662, dogs in SU 1663, and cows in SUs 573, 3171, and 1361 (Table 3). Animals from other SUs are represented in a partial and fragmented state, with only a few remains per individual (Appendix A). Nevertheless, when assessed globally, bones from all anatomical portions have been identified (Table 5 and Table 6).

For the Bell Beaker phase (Table 5, Figure 5), cattle’s and caprines’ cranial elements are prevalent, reflected in the abundance of dental pieces and fragments of cow horns and skulls. The second most represented anatomical section is axial bones, followed by appendicular elements. In contrast to this, there is a scarcity of pig axial bones (Table 6).

The skeletal profiles for the Middle Bronze Age follow a similar pattern to those observed for the Bell Beaker phase. There is a comprehensive representation of the skeleton, with a prevalence of cranial elements, followed by axial and appendicular bones (Table 7 and Figure 6). The better representation of cranial elements is once again reflected in the abundant dental pieces and fragments of skulls and horns (these elements tend to have a greater fragmentation). Finally, the low representation of long bones could be explained because the fragmentary remains of diaphyses can only be classified as undetermined remains from large- and small-sized animals (Table 7).

### 3.3. Taphonomical Analysis of Bone Surfaces

The bone surface analysis at Pista de Motos suggests the good preservation and minimal fragmentation of the faunal remains. The absence of weathering-induced cracking suggests rapid sedimentation and limited exposure over time. Additionally, there is no observed pigmentation from manganese or calcareous concretions. The main osteological alterations at this site are the modifications caused by biological agents, particularly carnivores, like dogs.

These carnivore marks, classified as tooth marks, including pits and scores, are present across almost all taxa (Figure 6), all anatomical sections (Table 8), and each chronological period (Table 9). In larger animals (i.e., cows and horses), the bones richer in fat, such as the axial bones, upper limb elements, and epiphyses (Table 8), are the most affected by carnivore activity. Conversely, in smaller animals, most marks are located on the diaphyses of long bones, contributing to their fracturing and the collapse of epiphyses (Table 8). Notably, SUs with animals in anatomical connection, including complete bones, lack tooth marks.

Rodents also left traces in the osteological record, leaving tooth marks on five rabbit specimens found in SUs 2411 and 2641 and on three caprine remains from SUs 821, 841, and 2821.

Table 9 and Figure 7 summarise the anthropogenic marks in the faunal osteological record at the Pista de Motos. Among these, the most prominent ones are cut marks from skinning, disarticulation, and defleshing activities (Table 10). Although burnt bones are also present, these have not been abundantly found. Most units only exhibit a limited amount of superficially burnt bones with mild thermal alteration (see Appendix A). In only a few units, like SUs 351, 901, 962, 1101, 1651, 2227, 2652, and 2961, burnt bones are significant, but never exceeding 30 remains. The remaining SUs are not statistically representative, according to NISP.

The presence of cut marks in all taxa, in both domestic and wild species, suggests butchering for meat consumption. This implies that wild animals, including horses, deer, boars, hares, and rabbits, were likely hunted. Similarly, all domestic animals, including dogs, display evidence of having been exploited for meat (Table 9). Disarticulation marks are characterised by cuts breaking the bone with transverse incisions and are observed especially on the humerus and femur metaphyses (refer to Figure 8). Additionally, disarticulation incisions resulting from fracturing jaws to access the tongue are present. These are identified through transverse cuts on the mandibular ramus, at the level of M1, and beneath the mandibular condyles (refer to Figure 9), indicating a potential facilitation of the butchering process.

As mentioned above, there is a clear preference for meat consumption in exploiting animal resources for food. This is further supported by the cut marks suggesting filleting and defleshing. Conversely, bone marrow does not appear to have been relevant. Besides many long bones being complete, the fractured ones result from processes unrelated to marrow consumption. On the one hand, there are fractures caused by carnivores’ gnawing bones, which can lead to the collapse of the epiphyses or, in some cases, the formation of cylinders. On the other, as already stated, some bones were fractured by humans, who cut through them to facilitate dismemberment and disarticulation (e.g., Figure 8).

In the exploitation of larger animals (e.g., cows), it has been observed that at the site, their long bones do not usually fragment unless they are disarticulated, which was only achieved through specific cuts (Figure 8 and Figure 9). Fractures on the skull’s base suggest the consumption of this animal’s brain. Access to the tongue is indicated by the disarticulation of the mandible, which is presented by cuts on the condyles and fractures at the M1 level, as indicated above. The consumption of the limbs, indicated by their disarticulation, is observed in various manners.

To access the forelimbs, on some occasions, the scapula was separated from the humerus by cutting the tendon. In other cases, there was a cut at the distal end of the scapula, resulting in a transversal fracture pattern at the articular end. The humerus was disarticulated from the radius–ulna by cutting tendons or through the transverse fracture of the humerus distal metadiaphysis (Figure 8). Evidence for the radius, carpal, and metacarpal access is limited. The metacarpus often does not fracture; in others, the radius breaks, transversally or longitudinally, at its distal metadiaphysis. To consume the hindquarters, these would have been disarticulated from the pelvis by the acetabulum in its connection with the femur. Alternatively, the femur was also cut through the proximal diaphysis. The disarticulation of the femur with the tibia occurred in various ways, creating longitudinal or transversal cuts in the metadiaphysis or distal epiphysis of the femur. Regarding the junction of the tibia with the metapods, cut marks associated with disarticulation have only been found on the talus.

Small animals like sheep, goats, and pigs show less fracture compared with larger animals, and in many cases, the marks are associated with dismemberment. For instance, transversely fractured caprines’ tibias are around the tibial crest and the distal metadiaphysis. For other elements, like the humerus, similar patterns to those described in the cow have been observed (see Figure 7). Other bones, like the metapodials or the radius, do not display evidence of intentional anthropic fracture.

Another hypothetical use for the faunal record at Pista de Motos is their utilisation in symbolic-ritual contexts. This would be suggested by the deposits with articulated fauna in SUs 857, 1018, 1666, 1662, 1663, 1664, 2692, 573, 3171, and 1361. In these, animals are complete, and their bones are not fractured, proving that their marrow was not extracted either. Only one cow from SU 1361 presents some cut marks associated with defleshing.

## 4. Discussion

The zooarchaeological and taphonomic analyses conducted at Pista de Motos have provided insights into various occupations, particularly in the Bronze Age. This period reveals two distinct phases characterised by Bell Beaker and Procogotas ceramics. Both the Bell Beaker and Middle Bronze Age periods share a similar faunal composition, defined by a predominance of bovine and caprine livestock, including mixed flocks of sheep and goats. These are followed by pigs and dogs, still within the domestic fauna. Wild animals are less represented, featuring remains of birds, Iberian ibex, deer, horses, wild boars, and some lagomorphs.

This zooarchaeological and taphonomic study revealed that the site’s fauna served both economic and symbolic purposes. Some animals were processed for their meat, while others were deposited entirely without manipulation. Lastly, certain remains are arbitrarily associated with human burials. Within domestic species, all animals except pigs are primarily represented by adults, likely related to the economic uses of the species at the site, a topic we will discuss further below.

Following the dual functionality of animals at the site, we aim to discern each species’ potential roles in life and death during the Bronze Age.

Starting with cattle, the most representative taxa, there appears to be an evolution of the exploitation pattern from the Bell Beaker to the Middle Bronze Age. The high percentage of infant and juvenile individuals (50% of the population) during the Bell Beaker Bronze Age (Figure 2) suggests that cattle were intended for meat consumption during this period. Conversely, during the Middle Bronze Age (Appendix A), an increasing representation of adult individuals and the possible presence of oxen suggest they might have also served other functions. Cows could have been used for milk production and oxen for labour. Cut marks on their bones also suggest continued use as meat resources post mortem. Unlike the Bell Beaker period, bovine food exploitation during the Middle Bronze Age appears less intensive, as suggested by limited bone fracturing.

Caprines, represented by sheep and goats, constitute the second most representative species. Similar to cattle, we detect an evolution of exploitation over time. The increase in adult individuals during the Middle Bronze Age (Figure 3) might be linked to the emergence of dairy or textile uses (i.e., wool exploitation). Ceramic objects like loom weights and cheese-making vessels further support this hypothesis. Besides this chronological differentiation, sheep and goats exhibit slightly different mortality patterns. Table 2 and Figure 2 show that goats display more infant–juvenile individuals and a broader mortality pattern during the Middle Bronze Age. Table 9 additionally indicates that goats’ remains display more cut marks than sheep’s. This suggests that while goats might have been more associated with year-round meat consumption, sheep were likely more utilised for wool production. Nevertheless, both species were exploited for meat once they were no longer functional in life (Table 9 and Table 10).

Pigs are the third most representative species at the site. Their mortality patterns and taphonomic analysis confirm that these were primarily kept for their meat.

Dogs appear to have served two primary purposes. The first is associated with hunting, as suggested by at least two medium-sized dogs, which measured 45 cm in height at the withers (Appendix A). The second function might have been associated with herding or rodent hunting, as the small-sized dog remains suggests. This type of dog was commonly used in rural areas for rodent hunting or to accompany livestock and guard against potential predator attacks. Examples of mouse-hunting dogs are rare in the literature, with most research primarily focusing on dogs as ungulate hunters. Yet, some authors highlight their proficiency as rodent hunters [72,73,74,75,76,77,78,79], including some specialised breeds [80,81,82]. They additionally emphasise how these capabilities might deplete the hunting resources of other carnivores, leading to their extinction [74,75]. Finally, similar to other domestic species, they were also used for meat consumption after their life cycle ended, as evidenced by certain cut marks (Table 10).

Among wild animals, only herbivores have been identified, all showing evidence of being used for meat consumption. However, their low frequencies, constituting less than 10% of the MNI in both the Bell Beaker period and the Middle Bronze Age, suggest that hunting was not a significant activity at Pista de Motos during the Bronze Age.

The patterns at Pista de Motos align with other Bronze Age sites documented in the interior of the Iberian Peninsula, specifically the Tagus Middle Valley (Table 10), where caprines and bovines also prevail. This can be observed in sites, among others, such as Matillas [41], Espinillo [37], Caserío de Perales [83], Velilla y Merinas [84], and Angosta de los Mancebos [33] (Table 11). Exceptions are noted at Loma del Lomo [83] and Barranco del Herrero [25], where suids are more prominent. The prevalence of pigs, particularly at Loma del Lomo, might be attributed to its geographical location, a mountainous area near Cogolludo in Guadalajara (see Figure 10). The wild fauna profile at Pista de Motos also follows the general trend of continuously reducing hunted species since the Neolithic [32]. Beyond the similar faunal profiles, Pista de Motos also shares the characteristic silo and pit features widespread in the Middle Tagus Valley (refer to Figure 10). These are even present in sites like La Loma del Lomo or Barranco del Herrero, where pigs were more abundant.

The main difference between Pista de Motos and other sites is their more representative faunal sample size. Additionally, it features entire buried animals, thus complementing the more widespread economic activities with a symbolic-ritual role. Similar symbolic-ritual observations have been made in other regional sites such as Camino de las Yeseras [22], la Fábrica de Ladrillos [24], Aldovea [32], and other unpublished locations like Humanejos or Barrio del Castillo. Still, these sites are in the early stages of research, and in many cases, the faunal representation has not been published yet.

After examining the fauna relationships within the regional context of Pista de Motos, a more detailed analysis of the bone assemblage at the site is now undertaken. The bone accumulation at Pista de Motos is characterised by two types: one associated with economic animal utilisation, emphasising cattle, goats, and pigs (Table 2 and Table 3), and the other linked to a symbolic treatment of the animals (left column in Table 4), with whole individuals buried without apparent additional exploitation.

Table 12 compares the Minimum Number of Individuals (MNI) for both accumulations, distinguishing between ‘Domestic’ for economic use and ‘Symbolic’ for symbolic treatment. During the Middle Bronze Age, symbolic assemblages featured cows as the most abundant animal, representing 39% compared with 15.7% in domestic ones. Caprines account for 22% of the MNI in symbolic deposits, contrasting with 5% in domestic contexts. Dogs and pigs each represent 11% of the MNI in the symbolic deposits, similar to the frequencies observed in the domestic assemblages. Finally, ravens, owls, and the wild Iberian ibex also have a higher percentage representation in the symbolic deposits than in domestic contexts.

Besides the differential representation of species in the deposits, differences are also observed in the individuals’ mortality patterns in both types of assemblages. In symbolic deposits, there was a preference for sacrificing infant or juvenile individuals in all taxa except dogs (Table 12). For instance, cows exhibit 43% non-adult individuals in the symbolic deposits against 19% in domestic assemblages (Figure 2, Table 12). The same trend is noted in goats and pigs. The findings from Pista de Motos align with observations from other faunal assemblages found at various sites. Bovids are the most represented animals, often showing the prominence of non-adult individuals. For instance, at Loma del Lomo de Cogollado (Guadalajara), in the Late Bronze Age layers, a juvenile bovid was associated with a young goat [83]. Similarly, several juvenile individuals were discovered at Camino de las Yeseras [90]. Thus, this pattern is consistent across multiple Bronze Age sites in the region. [15].

A particularity among the recovered symbolic fauna at Pista de Motos is the absence of carnivore alterations on the bones. This contrasts with the remains from the ‘Domestic’ units, where all taxa display both tooth and cut marks (Figure 6). This distinct treatment is exceptional, as once the animals were deposited, they were also buried, preventing the manipulation by external agents such as dogs. This special handling has also been observed in sites like Las Matillas [41], where evidence of dog and cow skulls was found, as well as at Tejar del Sastre [91], Perales del Río [87], La Dehesa [92], Getafe [93], and Camino de las Yeseras [90]. Therefore, the symbolic depositions at Pista de Motos also align with several other Bronze Age sites in the inland [15,79] and other regions within the Iberian Peninsula [15,94].

## 5. Conclusions

Pista de Motos stands out as a valuable addition to the zooarchaeological samples from the Iberian Peninsula inland. It is one of the most significant faunal collections known to date, offering essential information about exploiting livestock resources during the Bronze Age period. Nevertheless, despite the significant contribution of these new data and observations, research for this site and this period should continue. Further approaches should involve analyses targeting specific questions. For example, a more robust seasonal analysis could be achieved using cement chronology or dental microwear techniques. Herd mobility could be analysed through stable isotopic analysis. New taphonomic studies could clarify whether metal or lithic tools were used for the cut marks analysed here. Finally, more sites in the region should be researched to address the evolution of animal exploitation across the Middle Tagus Valley from the Chalcolithic to the Roman Age.

## Figures and Tables

**Figure 1 animals-14-00413-f001:**
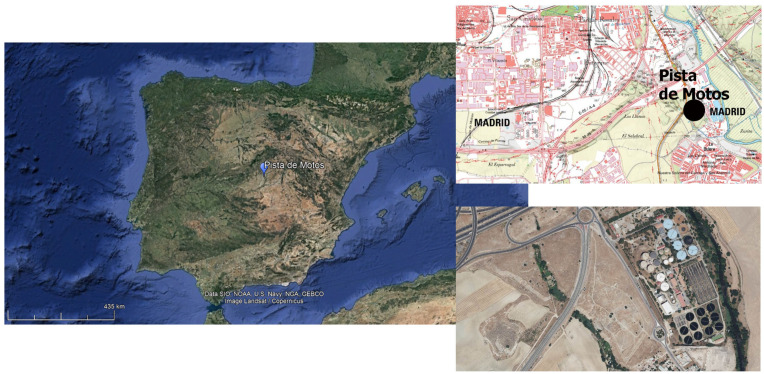
Pista de Motos’ (Villaverde, Madrid) location. The image on the left shows the regional location of the site in a modified image extracted from Google Earth. The upper and lower right images display a more refined site location.

**Figure 3 animals-14-00413-f003:**
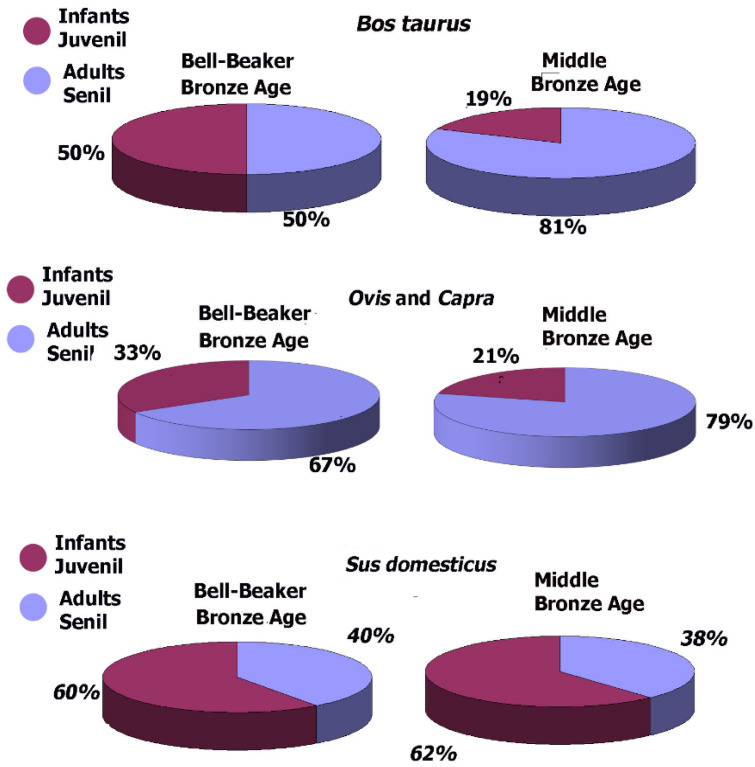
Mortality patterns are divided between immature and adults. Juveniles and infants are the most representative among the most prevalent domestic fauna taxa.

**Figure 4 animals-14-00413-f004:**
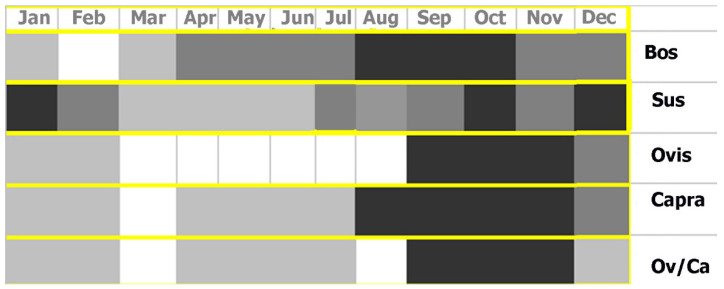
Seasonal death estimates for the Middle Bronze Age cows, pigs, and goats. According to MNI, the colour ranges indicate when more death episodes occur. For more information, see Appendix A.

**Figure 5 animals-14-00413-f005:**
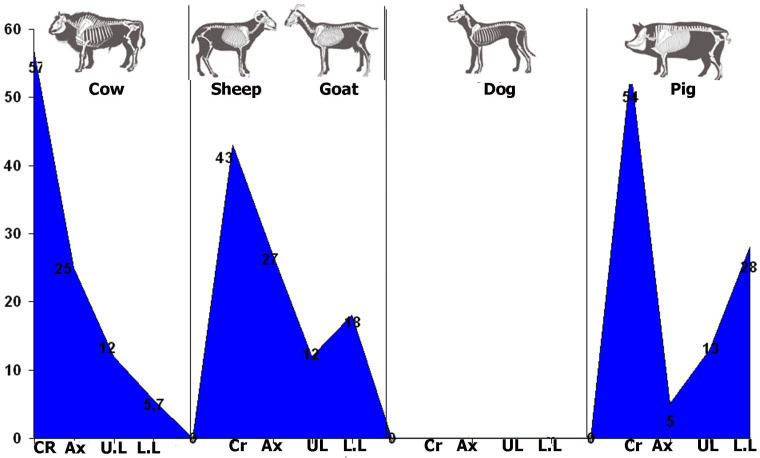
Frequencies of skeletal profiles by sections for most prominent animals during the Bell Beaker period. Following Table 6, CR: Cranial; Ax: Axial; U.L.: Upper Limbs; L.L.: Lower Limbs.

**Figure 6 animals-14-00413-f006:**
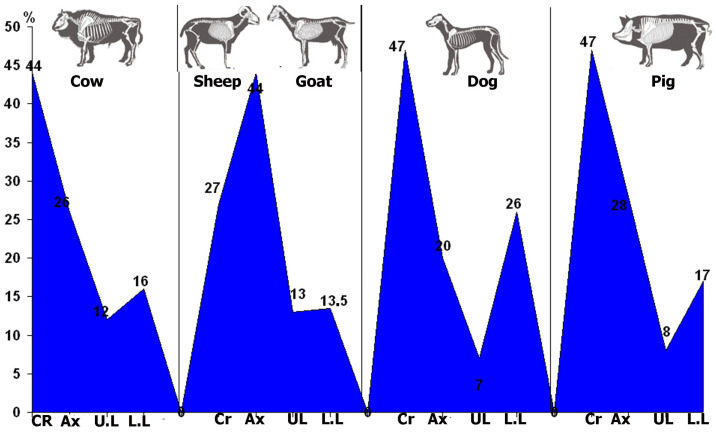
Frequencies of skeletal profiles by sections for the most prominent animals during the Middle Bronze Age. Following Table 7, CR: Cranial; Ax: Axial; U.L.: Upper Limbs; L.L.: Lower Limbs.

**Figure 7 animals-14-00413-f007:**
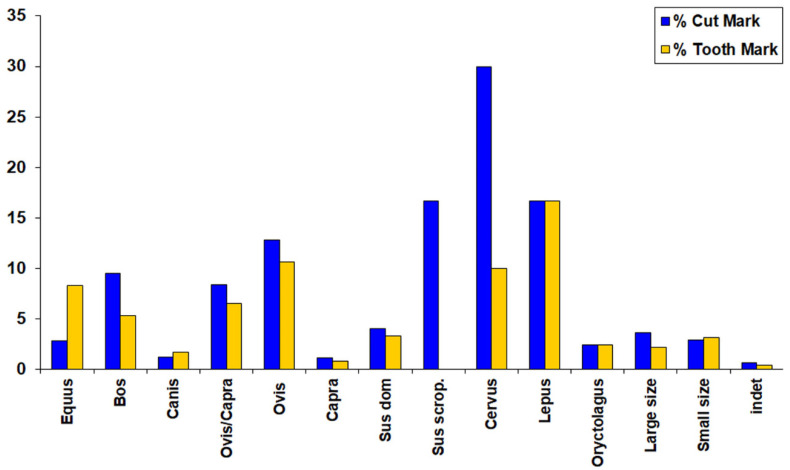
Cut marks and carnivores’ marks frequencies at Pista de Motos.

**Figure 8 animals-14-00413-f008:**
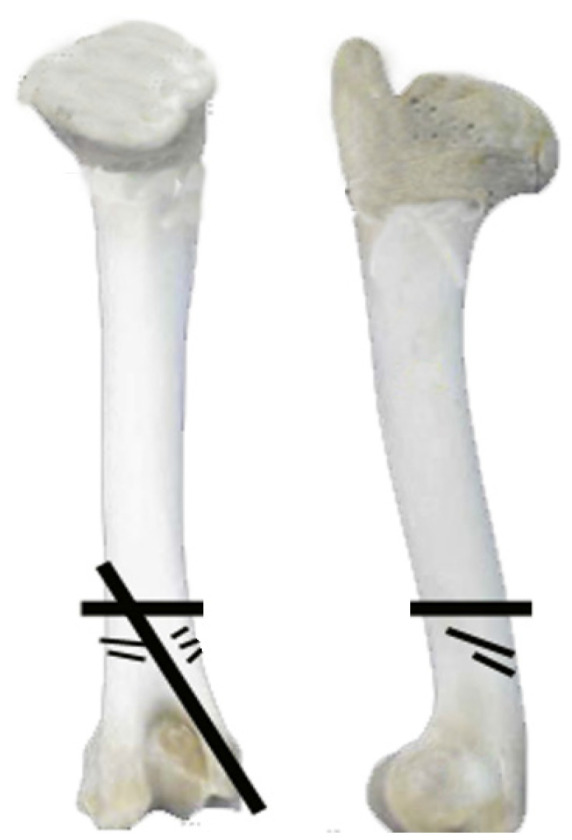
Disarticulation marks (fine lines) and fracture patterns (gross lines) in *Bos taurus* long bones. This is exemplified on a humerus.

**Figure 9 animals-14-00413-f009:**
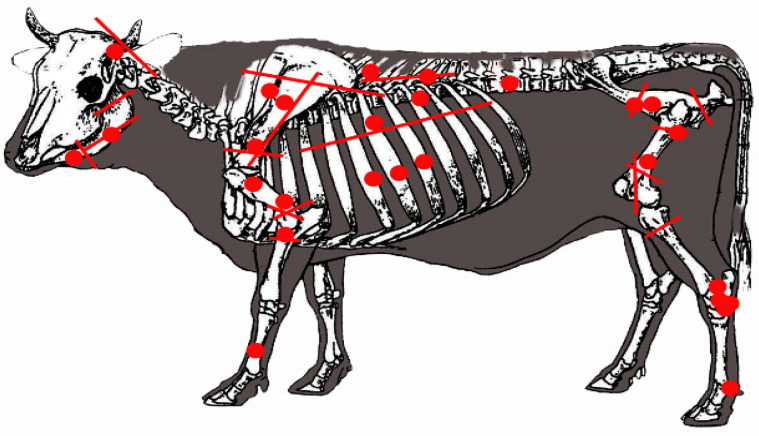
Disarticulation patterns and cut marks are marked with red lines and circles on a cow. The model was taken from [62].

**Figure 10 animals-14-00413-f010:**
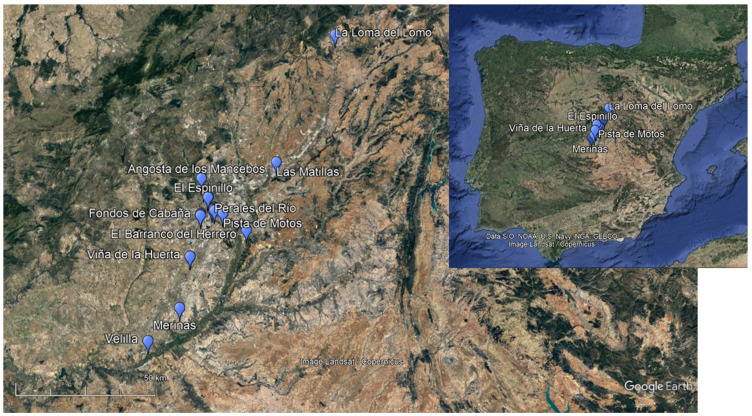
The location of the Bronze Age archaeological sites with faunal remains that are presented in Table 11.

**Table 1 animals-14-00413-t001:** The uncalibrated and modelled dates and boundaries for Pista de Motos’ Bronze Age.

Phases/Samples	Uncalibrated Dates BP [21]	Modelled Date (95.4%) cal BP
End Middle Bronze Age		3384–2933
SU1660	3090 ± 24	3381–3241
SU2581	3161 ± 27	3450–3343
SU962	3269 ± 21	3558–3402
Start Middle Bronze Age		3901–3407
End Bell Beaker		4219–3691
SU2417	3764 ± 25	4232–4001
SU1251	3768 ± 31	4234–4001
Start Bell Beaker		4546–4011

**Table 2 animals-14-00413-t002:** Faunal representation of Pista de Motos according to NISP and %NISP. This Table only shows the classified samples for the Bronze Age period.

NISP	Neolithic	Bell Beaker Bronze Age	%	Middle Bronze Age	%	Iron Age	Visigoths	Islamic	Prehistoric	Total
*Bos taurus*	3	263	27.4	1293	18.9	2	15		204	1779
*Ovis aries*		17	1.8	53	0.8	7	3		19	99
*Capra hircus*	1	8	0.8	333	4.9		1		6	523
*Ovis/Capra*	6	103	10.7	969	14.2	5	15	1	107	1206
All *Ovis* and *Capra*	7	128	13.3	1529	22.4	12	19	1	132	1828
*Canis famil.*		1	0.1	476	7.0				64	541
*Sus domest.*		39	4.1	604	8.8		12		21	654
*Sus scrofa*		2	0.2	9	0.1					11
*Equus ferus*		21	2.2	13	0.2				6	40
*Capra pyrenaica*				174	2.5					
*Cervus elaphus*		1	0.1	9	0.1					10
*Oryct. cuniculus*		32	3.3	41	0.6	1	1		8	83
*Lepus* sp.			0.0	5	0.1				1	6
*Corvus* sp.			0.0	45	0.7					45
*Tyto alba*			0.0	28	0.4					28
*Alectoris rufa*		4	0.4							4
Birds indet				1	0.0					1
*Bufo bufo*						18				18
Large A. Size	8	302	31.5	1335	19.5	6	5	1	169	1824
Small A.Size	18	147	15.3	991	14.5	10	32	4	173	1372
indet.		19	2.0	469	6.9		5		30	522
Total	36	959		6849		49	89	6	809	8766

**Table 3 animals-14-00413-t003:** Taxonomic representation of Pista de Motos following the MNI (Minimum Number of Individuals). Age patterns are described for the samples where S (senile), A (adult), J (juvenile), I (infant). *Bufo bufo* has been excluded from this table because it is an allochthonous remain.

MNI	Neolithic	Bell Beaker Bronze Age			Middle Bronze Age			Iron Age	Visigoths	Islamic
	S-A-J-I	S-A-J-I	MNI	%	S-A-J-I	MNI	%	S-A-J-I	S-A-J-I	S-A-J-I
*Bos taurus*	0/1/0/0	0/3/2/1	6	16.7	2/11/2/1	16	15.7	0/1/0/0	1/1/0/0	
*Ovis aries*		1/2/0/0	3	8.3	1/4/0/0	5	4.9	0/1/0/0	0/1/0/0	
*Capra hircus*	0/1/0/0	1-1-0-0	2	5.6	2/6/1/3	12	11.8		0/1/0/0	
*Ovis/Capra*	0/1/0/0	1-2-1-1	5	13.9	1/12/1/1	15	14.7	0/1/0/0	1/1/0/0	0/1/0/0
All *Ovis* and *Capra*		1/3/1/1	6	16.7	3/12/1/3	19	18.6	0/1/0/0	1/1/0/0	0/1/0/0
*Canis familiaris*		0/1/0/0	1	2.8	0/10/0/0	10	9.8			
*Sus domesticus*		0/2/1/2	5	13.9	1/4/5/3	13	12.7		0/1/0/0	
*Sus scrofa*		0/1/0/0	1	2.8	0/2/0/0	2	2.0			
*Equus ferus*		0/2/0/0	2	5.6	0/1/0/0	1	1.0			
*Cervus elaphus*					0/1/0/0	1	1.0			
*Capra pyrenaica*					0/0/1/0	1	1.0			
*Oryct. cuniculus*		0/4/0/0	4	11.1	0/3/0/1	4	3.9	0/1/0/0	0/1/0/0	
*Lepus* sp.					0/1/0/0	1	1.0			
*Corvus* sp.					0/1/0/0	1	1.0			
*Tyto alba*					0/1/0/0	1	1.0			
*Alectoris rufa*		0/1/0/0	1	2.8						

**Table 4 animals-14-00413-t004:** Summary of the main SUs with the most significant faunal samples, divided into those with fauna deposits where animals were deposited completely (C) or partially (P) and in anatomical connection. MNI: Minimum Number Individual. The chronological assignation is Pre: Prehistoric, MBA: Middle Bronze Age, BSB: Bell Beaker Bronze. Age Patterns are Ad: Adult, Juv: Juvenile, Inf: infantile. The second column only lists the most representative taxa of the SU. For more detail on the taxonomic representation of the SUs, refer to Appendix A.

SU with Animals as a Deposit. Complete (C) or Partial (P) Named “Deposit”	SU with Bones Sample > 100 NISP	SU with Fauna and Human Inhumations
1018 (Dog. MNI = 1 ad) (C) Pre 1666 (Raven, MNI = 1 ad) (C) MBA 1361 (Cow, MNI = 3ad) (1C-1P-1P) MBA 1361 (Barn Owl, MNI = 1ad) (C) MBA 1661 (Dog, MNI = 1ad) (P) MBA 1662 (Pig, MNI = 1 Inf) (C) MBA 1663 (Dog, MNI = 1ad) (C) MBA 1664 (Goat, MNI = 1 Inf) (C) MBA 2692 (Goat, MNI = 1 ad) (C) MBA 2693 (Goat, MNI = 1 Inf) (P) MBA 3171 (Cow, MNI = 2 juv) (C-P) MBA 857 (iber.Ibex–1juv) (C)—and goat –ad-(P) MBA 573 (Cow, MNI= 2 juv and ad) (C-P) BSB 981 (Cow, MNI = 1ad) (P) MBA 2371 (Pig MNI = 1 inf (P) MBA	1251 *Ovis/Capra* 1331 Pig and *Ovis/Capra* 1336 *Ovis/Capra* 120 Pig 2201 *Ovis/Capra* 2371 *Ovis/Capra*) 2391 Cow 2417 *Ovis/Capra* 3001 Pig, Cow and *Ovis/Capra* 3021 Cow and *Ovis/Capra* 3036 *Ovis/Capra* 481 Cow 511 Cow 856 Cow 571 Cow 573 Cow	1401 MBA 2227 Visig. 2565 Islam. 963 y 962 MBA

**Table 5 animals-14-00413-t005:** MNI extracted from the sum of each specific unit observed in Appendix A.

MNI	*Bos*	*Equus*	*Ovis*	*Capra*	*Ovis/Capra*	*Canis*	*Sus d.*	*Sus sc*	*Oryctol.*	*Lepus*
Neolithic	2			1	2					
Bell Beaker Bronze Age	17	5	5	5	9	2	6	1	5	
Middle Bronze Age	144	6	22	51	126	40	61	2	25	
Iron Age II	2		2		2	1			1	
Visigoths	3		1	1	3		1		1	
Islamic Age					1					
Prehistoric	27	3	9	1	20	1	5		3	1

**Table 6 animals-14-00413-t006:** Macrovertebrates’ skeletal profiles for the Bell Beaker period.

	*Bos taurus*	%	*Ovis aries*	*Capra hircus*	*Ovis/Capra*	*Ovis* and *Capra.* Total	%	*Equus*	*Sus*	%
Horn	41	15.6					0.0			0.0
Craneal	34	12.9			5	5	3.9			0.0
Maxillar	1	0.4					0.0		4	10.3
Mandible	13	4.9	2	1	7	10	7.8		2	5.1
Teeth	62	23.6	13	9	18	40	31.3		15	38.5
Vertebrae	33	12.5			8	8	6.3	3		0.0
Rib	21	8.0			20	20	15.6	1		0.0
Scapule	5	1.9	1		6	7	5.5	1		0.0
Humerus	6	2.3			2	2	1.6	2		0.0
Radio	6	2.3			6	6	4.7	1		0.0
Ulna	6	2.3					0.0	1	2	5.1
Metacarpal	4	1.5			3	3	2.3			0.0
Carpal		0.0			1	1	0.8			0.0
Pelvis	7	2.7					0.0	2	2	5.1
Femur	6	2.3			1	1	0.8	2	3	7.7
Tibiae	7	2.7			6	6	4.7	3		0.0
Metatarsal	3	1.1		1	4	5	3.9	1		0.0
Metapodial	3	1.1			1	1	0.8		6	15.4
Talus	2	0.8	1		8	9	7.0			0.0
Calcaneus		0.0		1	1	2	1.6	1		0.0
Phalange	3	1.1			2	2	1.6	3	5	12.8
Total	263	100	17	12	99	128	100.2	21	39	100

**Table 7 animals-14-00413-t007:** Macrovertebrates’ skeletal profiles for the Middle Bronze Age.

	*Sus scro*	*Cervus*	*Lepus*	*Bos*	%	*Ovis*	*Capra*	*Ov/Cap*	All *Ovis* and *Capra.*	%	*Equus*	*Canis*	*Sus*	%	*Oryc.*	Indet Large Size	Indet Lower Size
Horn				40	3.1		8		8	0.5							
Cranial				222	17.2		1	67	68	4.4		6	60	10.4		59	11
Maxillar	1			25	1.9		8	9	17	1.1		16	12	2.1			
Mandíble				52	4.0	3	15	34	52	3.4	1	21	19	3.3	4	13	2
Teeth	5			227	17.6	23	140	98	261	17.0	3	211	180	31.3		5	
Vertebrae				125	9.7	1	182	58	241	15.7		49	64	11.1	1	21	12
Rib				189	14.6	0	103	280	383	25.0	3	51	85	14.8		45	7
Scapule			1	18	1.4	2	8	21	31	2.0		2	4	0.7	1	3	1
Humerus				37	2.9	4	12	35	51	3.4	4	4	10	1.7	4	22	5
Radio	1	1	2	55	4.3	6	26	13	45	2.9		8	10	1.7	6		
Ulna	1			22	1.7		10	10	20	1.3		8	9	1.6	2		
Metacarpal		1		22	1.7	5	5	3	13	0.8		10	8	1.4			
Carpal				49	3.8		30		30	2.0		20	10	1.7		2	
Pelvis		1		8	0.6	0	9	17	26	1.7		4	7	1.2	5	5	
Femur		2		16	1.2	1	7	14	22	1.4		4	10	1.7	2	1	1
Tibia	1	1	2	43	3.3	2	6	47	55	3.6	1	8	7	1.2	8	7	7
Fibula					0.0					0.0		2	0	0.0			
Patella				3	0.2			1	1	0.1		4	2	0.3			
Metatarsal		2		20	1.5	3	8	25	36	2.3		10		0.0			
Metapodial		1		9	0.7			5	5	0.3	1	5	14	2.4	7	2	4
Talus	2			15	1.2	2	8		10	0.7		4		0.0			
Calcaneus		1		13	1.0	1	6	2	15	1.0		5	2	0.3	1		
Tarsal				1	0.1		18		18	1.2		12	6	1.0			
Sesamoid				10	0.8			1	1	0.1				0.0			
Phalange				71	5.5		72	7	79	5.2		72	57	9.9			
Indet Shafts								41	41	2.7						1140	838
Total	11	10	5	1292	100	53	694	788	1532		13	536	576		41	1325	988

**Table 8 animals-14-00413-t008:** Tooth marks are classified by sections: Cranial, Axial, Epiphysis, and Diaphysis. The tooth mark frequencies are only shown for the most representative taxa. For more information, see Appendix A.

Anatomical Section	*Bos*	*%*	*Equus*	*Capra*	*Ov/Ca*	*Ovis*	*Ovis/Capra*	*%*	*Canis*	*Sus*	Large Size	Small Size
Cranial	1	1.3		1	6						2	
Axial	21	26.6	1	2	11	1	14	19.7	2	4	10	1
Ep-Dist	6	7.6	1	1	1		2	2.8		2	2	
Ep-Prox	13	16.5			3	2	5	7.0				
Epiphysis	19	24.1	1	1	4	2	7	9.9		2	2	
Compact Bones	7	8.9								1		
Shafts	31	39.2	1	3	35	2	40	56.3	4	8	26	42
Total	79		3	7	56	5	71		6	15	40	43

**Table 9 animals-14-00413-t009:** NISP with cut and tooth marks observed at Pista de Motos during the Bronze Age.

NISP	Cut Mark		Tooth Mark	
	Bell Beaker Bronze Age	Middle Bronze Age	Bell Beaker Bronze Age	Middle Bronze Age
*Bos taurus*	10	108	7	70
*Equus ferus*	0	0	2	1
*Capra hircus*	0	9	0	7
*Ovis aries*	0	6	0	4
*Ovis/capra*	8	53	5	45
*Sus domestic*	0	16		12
*Oryct. cuniculus*	1	1	0	2
*Canis familiaris*		4	1	5
*Cervus elaphus*				1
*Sus scrofa*		1	1	
Indet small animal size	5	27	1	36
Indet large animal size	9	51	5	28
Total	33	276	22	214

**Table 10 animals-14-00413-t010:** The location and function of cut marks at Pista de Motos. Axial (vertebrae, ribs, scapule, and pelvis), Cranial (mandible, maxillar, and craneal), Shafts (diaphysis of humerus, femur, tibia, radius, and metapodial), Compact Bones (tarsal, carpal, sesamoids and phalanges).

	Axial	Cranial	Shafts	Ep-Dist	Epiph.	Ep-Prox	Compact Bones	Total
*Equus*	0	0	1	0		0	0	1
*Bos*	45	10	31	6		18	31	141
*Oryctolagus*	0	0	1	1		0	0	2
*Ovis/Capra*	39	4	26	2		0	1	72
*Ovis*	2	0	2	1		0	1	6
*Capra*	3	1	4	0		1	0	9
*Canis*	2	2	0	0			0	4
*Sus*	3	3	9	2		1	0	18
*Cervus*	0		0	2		0	1	3
*Lepus*	0		0	1				1
Large Siez	18	5	41	0	1	0	0	65
Small Size	8	0	32				0	40
Indet.	2	0	1		0		0	3
Total	122	25	148	15	1	20	34	365
Functionality	Disarti-culation	Skinning	Filleting and Disarticulation	Disarti-culation	Disarti-culation	Disarti-culation	Skining and Disarticulation	

**Table 11 animals-14-00413-t011:** Taxonomic representation of the main Bronze Age sites with faunal representation in the Middle Tagus Valley.

	Matillas	Espinillo Chalcol.Bronze	Loma del Lomo	Espinillo Bronze	Angosta Mancebos	Barr. Del Herrero	Fondos Cabaña Getafe	Fábrica de Ladrillos	Caserío Perales	Perales del Río	Velilla I	Arenero Soto	Merinas	Viña de la Huerta I
References	[41]	[37]	[83]	[37]	[33]	[25]	[85]	[24]	[86]	[87]	[84]	[88]	[84]	[89]
	NISP	NISP	NISP	NISP	NISP	NISP	NISP	NISP	NISP	NR	NISP	NISP	NISP	NISP
*Equus ferus*		8	8	3			6	7	11	2		6		
*Bos taurus*	78	70	190	20		5	118	697	23	30	67	118	7	79
*Capra hircus*						2	5	9		1		12		3
*Ovis aries*	3					5	12	9		3		6		27
*Ovis/Capra*	91	83	296	25	21	10	258	256	10	54	40	258	10	42
*Canis familiaris*	6	144	44		1		22	12	6	7	9	22		3
*Sus domesticus*	18	27		7	2	16	3	47		8		3		5
*Sus scrofa*	3						4					4		
*Sus* sp.			144				42		2			42		
*Cervus elaphus*	26	4	12	4	2	1	22	4		3	2	22	1	1
*Capreolus capre*			1											
*Vulpes vulpes*		1					1					1		
*Oryctol. cunicul.*		16	15	2			4	62		13		4		
*Lepus* sp.			3		1		8	10			1	8		
Leporidae indet							59	4				59		
Large size						12			42		65		9	303
Intermediate size									5		40			
Small size						20			59				8	161
Carnivores								8						
Indet								1824		98			27	84
Total	225	353	713	61	27	71	564	2949	158	220	266	565	62	708

**Table 12 animals-14-00413-t012:** The main characteristics of the faunal assemblages found in the MNI domestic bone assemblage of Pista de Motos and the MNI of symbolic deposits bone assemblage.

	Domestic Bone Assemblages (See Data of Table 3)		Symbolic Deposits Bone Assemblages (See Data of Table 4)	
	%MNI	% Infant or Juvenil	%MNI	% Infant or Juvenil
*Bos taurus*	15.7%	19%	39%	43%
*Capra hircus*	5%	33%	22%	50%
*Sus domesticus*	12.7%	69%	11%	100%
*Canis familaris*	9.8%	0%	11%	0%
*Capra pyrenaica*	2.5%	100%	5.5%	100%
Birds	1.1%	0%	11%	0%
Skeletal profiles	Skeletal profiles compensated with bones of all anatomical parts		All individual are complete and articulated	
Taphonomy	Human and carnivore activity		No human or carnivore activity	

## Data Availability

The data are contained within Appendix A and the samples are in the Regional Archaeological Museum of Madrid.

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
