# Peer review of "New Evidence for the Bronze Age Zooarchaeology in the Inland Area of the Iberian Peninsula through the Analysis of Pista de Motos (Villaverde Bajo, Madrid)"

_animals, 2024, doi:10.3390/ani14030413_

Round 1

Reviewer 1 Report

Comments and Suggestions for Authors

The manuscript titled "New Evidence for Bronze Age Zooarchaeology in the Iberian Peninsula Interior through the Analysis of Pista de Motos 3 (Villaverde Bajo, Madrid)" explores the faunal remains discovered at a Bronze Age site situated in the interior of the Iberian Peninsula. Given the limited understanding of human-animal interactions in this region during this period, this manuscript contributes significantly to Bronze Age archaeology. The faunal remains discussed in the manuscript are associated with subsistence and economic activities (meat, milk, wool, and animal labor force), as well as symbolic-ritual practices, as indicated by complete animal burials discovered in certain units.

However, improving the manuscript with additional figures could clarify the features of the site, which are currently only described. Including a site map with markers indicating the locations of silos/pits housing deposits of complete animal skeletons would provide spatial context. Additionally, supplementing the text with images of these complete skeletons, along with photographs featuring dentition used to determine the season of animal death and details of gnawing and cut marks on the bones, would improve the manuscript.

Minor suggestions

Line 119: Table 1 should be Table 2,

Line 196: Tables 1 and 2 should be Tables 2 and 3,

Line 123 and in tables: “Sus scropha” should be “Sus scrofa"

Line 197: "Thus, it mainly focused on exploiting herds of caprines and bovines, as well as pigs." Consider replacing "it" with "the economy," or "the inhabitants of the site," etc.

Line 212: Table 3 should be Figure 3

Table 3: "Canis familiares" should be corrected to "Canis familiaris"

2.12.0.0 2.12.0.0 Comments on the Quality of English Language

2.12.0.0 2.12.0.0

Author Response

Referer 1: The manuscript titled "New Evidence for Bronze Age Zooarchaeology in the Iberian Peninsula Interior through the Analysis of Pista de Motos 3 (Villaverde Bajo, Madrid)" explores the faunal remains discovered at a Bronze Age site situated in the interior of the Iberian Peninsula. Given the limited understanding of human-animal interactions in this region during this period, this manuscript contributes significantly to Bronze Age archaeology. The faunal remains discussed in the manuscript are associated with subsistence and economic activities (meat, milk, wool, and animal labor force), as well as symbolic-ritual practices, as indicated by complete animal burials discovered in certain units.

Answer:  Thank you very much for your evaluations. We appreciate your comments.

Referer 1: However, improving the manuscript with additional figures could clarify the features of the site, which are currently only described. Including a site map with markers indicating the locations of silos/pits housing deposits of complete animal skeletons would provide spatial context.

Answer: Thank you very much for this appreciation. If we had the map with the representation of all the locations of silos/pits, we would have included them in the manuscript. Unfortunately, we do not have this information at the moment.

Additionally, supplementing the text with images of these complete skeletons, along with photographs featuring dentition used to determine the season of animal death and details of gnawing and cut marks on the bones, would improve the manuscript.

Answer:  We agree with these comments. However, when we analysed the materials, we did not take pictures of the bone alterations nor of the occlusal alterations of the teeth. We cannot take the photos now because the materials are deposited in the regional archaeological museum, which is currently being renovated. Thus, it is not possible to access the materials at this time. This is the reason why we have not been able to include images of the materials.

Minor suggestions

Line 119: Table 1 should be Table 2,:

Answer: This has been corrected.

Line 196: Tables 1 and 2 should be Tables 2 and 3,

Answer: This has been corrected.

Line 123 and in tables: “Sus scropha” should be “Sus scrofa"

Answer:  This has been corrected.

Line 197: "Thus, it mainly focused on exploiting herds of caprines and bovines, as well as pigs." Consider replacing "it" with "the economy," or "the inhabitants of the site," etc.

Answer: This has been corrected.

Line 212: Table 3 should be Figure 3

Answer: This has been corrected.

Table 3: "Canis familiares" should be corrected to "Canis familiaris"

Answer: This has been corrected.

2.12.0.0 2.12.0.0

Comments on the Quality of English Language

Answer: This has been corrected. A researcher from the University of Oxford has reviewed the text.

Reviewer 2 Report

Comments and Suggestions for Authors

Dear authors,

Thank you so much for this manuscript - its content is much needed by those studying zooarchaeology. 

I am attaching the manuscript with comments throughout. I urge the authors to revise the English of the manuscript to improve the argument's presentation and discussion.

Also, please consider a more easy data presentation - some of the tables have a significant amount of data. 

Regards.

Comments on the Quality of English Language

The authors need to revise the English of the manuscript, as it will improve data presentation and overall interpretation and discussion.  

Author Response

Comments and Suggestions for Authors

Dear authors,

Thank you so much for this manuscript - its content is much needed by those studying zooarchaeology. 

I am attaching the manuscript with comments throughout. I urge the authors to revise the English of the manuscript to improve the argument's presentation and discussion.

Also, please consider a more easy data presentation - some of the tables have a significant amount of data. 

Regards.

 Answer: We really appreciate your comments, suggestions, and reviews in the attached document. We have revised the text with particular attention to our English writing.

Comments on the Quality of English Language

The authors need to revise the English of the manuscript, as it will improve data presentation and overall interpretation and discussion.  

Submission Date

16 December 2023

Date of this review

28 Dec 2023 23:22:51

According to the comments of PDf documment:

Referer 2: Line 42: better use the work - "inland"

Answer: We have followed the reviewer's comment and changed the term “interior” to “inland”.

Referer 2 Line 42-43 Also, It would benefit those not acquainted with the Iberian Peninsula a brief description

Answer: We have added the names of the administrative regions

Referer 2 Line 45: Deposits OR archaeological sites? Not all archaeological sites are deposits. Please clarify.

Answer: We have followed the reviewer's comment and clarified the sentence.

Referer 2 Line 56: Do you mean "scarce data/information" which have limited the research undertaken at /based with those sites, and consequently scientific advancements?

There will be the need to have the English revised by a proficient or native speaker

Answer: We have reviewed the text and changed the sentence to improve its expression and the meaning of what we want to say.

Referer 2 Line 61: Not the appropriate word.

Answer:  We have removed this word

Referer 2 Line 71: Not accurate English - please revise. 

Answer: we have revised and changed the sentence.

Referer 2 Line 81I think the aim was to say: "16 diferrent features found in 2717 sherds"?

Answer: we have revised and changed the line.

Referer 2 Line 82 and 84 Sentence should start with the name of the authors, not their reference number. 

Answer: We have corrected these references following the reviewer's comment.

Referer 2 Line 95-97. This, and previous sentence are confusing. Since there is a significant effort in dating the sate, a statement should also be added as to the benefits of such detailed chronology as opposed to wider chronologies

Answer: For us, it is essential to refer to previous absolute dates, as they are crucial to verify that we are working with materials from the Bronze Age. They are also helpful in referring to the only previously published paper on the site.

Referer 2 Line 107. The next sections are material and methods, not necessarily a discussion / state of the art on the use of domestic animals. Please rephrase this final sentence, or consider removing it.

Answer: We have eliminated this phrase because it really does not contribute anything to the text.

Referer 2 Line 110 Please provide a detailed description of all images

Answer: We have followed the reviewer’s suggestions and have completed the image’s caption.

Referer 2 Line 119 Specify where are in Table 1 silos and pits,  as the 1st column is a reference to samples / phases.

Answer: We have corrected this reference. It is actually Table 2.

Referer 2 Line 121 In the sense that it has o+more fragment of remains?.

Answer: We have corrected this reference. These are “bone fragments”

Referer 2 Line 122-124 This could be interpreted as a result already.

Answer: It could be. However, with this statement, we want to highlight that we will only focus on macrovertebrates in this study. This is why we have included this statement to clarify that we will only analyse the remains of macromammals.

Referer 2 Line 137 and line 146  When referring a specific paper, as in this case, please provide the author(s) then add the numerical referenc

Answer: We have corrected this reference following the reviewer's comment.

Referer 2 Line 147 Age mortality patterns, since we may also observe

Answer: This has been reviewed and corrected

Referer 2 Line 150: This sentence makes no sense .... I understand what the aim is, it it is not translated by the sentence: please revise.

Answer: We have revised and reduced this paragraph

Referer 2:  Line 158: Why use the word biological here?.

Answer: Because many researchers include anthropogenic alterations within biological alterations. We have simply followed the suggestions of other reviewers in other articles. But we believe that the sentence's meaning is understood if we exclude the term "biological", so we have eliminated this term in the phrase.

Referer 2 line 163-164.. This is a result of the observation, I would therefore suggest it to be included in the results.

Answer: This is true. We have moved this statement to results when discussing fracturing in the results section.

Referer 2 line 166. something is missing: profile? Analysis? Data?

Answer: We have revised and corrected.

Referer 2 line 176-184.This information should be linked with the detailed information on the anatomical location / bone element  that compose the overall sample. This data is explored bellow.

Answer:  We do not understand what the reviewer is referring to here. We have referred to the Tables where the information is, but at this stage of the results section, there is a better place to comment on the skeletal profiles since that part is addressed later.

Referer 2 line 197 Meaning?

Answer: We have revised and corrected this sentence.

Referer 2 line 203. How many fragments?  It would be interesting to also provide / make a specific reference to that data.

Answer: We have completed the specific reference.

Referer 2 line 204. This needs rephrasing. Also: why is it "notable"?

Answer: We have revised and corrected this sentence.

This is notable because it is significant that whole altered animals appear without human modifications.

Referer 2 line 206. The title needs to be explicit. Also, the % are not present in all cases - why?  Data presentation needs to be coherent and as complete as possible.

Answer: We have corrected the caption and included why percentages are not included in all cases. We show the % only in the material assigned to the Bronze Age.

Referer 2 line 208-209. The Table title should be explicit as to ist content. Any legends should be placed below the table, or below the table title. Also, data presentation needs to be coherent and as complete as possible. This is not the case in this Table and others - please revise. Inconsistencies create doubts, and/or misinterpretations.

Answer: We have reviewed the captions of all the Tables

In this paper, the table captions have been placed above the tables. I usually place the captions below, but in this journal, they are placed above.

Referer 2 line 222-Not sure this is the best word to use here, as sacrifice, in archaeological contexts may relate to rituals; but this articles as privileged the assumption of consumption as main cause for the presence of the many remains found.

Answer: We have changed the term for a more specific one

Referer 2 line 224-226 It is not clear how seasonality has been inferred. This needs more clarification.

Answer: We had explained it in the methods section. We state that by estimating the age in months and knowing the time of birth, a relative estimate can be made about the time of death.

Referer 2 line 238  Based on ??? more number of remains? This needs a better explanation.

Answer: We have modified the caption

Referer 2 line 241-244. It would have make more sense to present 1st the individual's in anatomical connection, and then its breakdown per fragments.

The section on Methods would benefit for having a paraphrase with an explanation on the results presentation.

Answer: In reality, we have followed the same scheme as in the methods, commenting on each section. In any case, to facilitate reading and following the comments of reviewer 3, we have divided the results into several sections.

Referer 2 line 241-244: In some sections we have US, in other SU, and EU ..... please provide information if different, or be consistent with data presentation

Answer: We have unified the terminology. Both are the same.

Referer 2 line 246-247 This needs detailed interpretation.

Answer:  We have added an explanation according to the reviewer

Referrer 2 line 262 Surely - this could be done for this paper. It would benefit the information, and contextual analysis tremendously.

Answer:  Indeed, as we show in the paper, this has been done for Table 5, the evidence of which is discussed later. We have written a small explanation to show these results at the end of this sentence.

Referrer 2 line 288-292 Please revise the table to provide consistency. US, SU, EU are used  ..... please provide a definition to each.

Answer: We have revised the table and provided greater coherence to the terminology.

Referrer 2 line 343-344 How was this tested?

Answer: The data on burnt bones can be observed in the supplementary file, and if the data from SF2 and SF 6 Table 9  are compared, the representativeness of the samples can be observed.

Referrer 2 line 343 Sf6 Table 8?

Answer:  It is SF 6 Table 9 not Table 8

Referrer 2 line 362-368 It would have been very interesting to have original photos from the cuts, etc.

Answer: We agree but could not take good-quality photos when we did the study. We cannot access the samples deposited in the Regional Archaeological Museum of the Community of Madrid. Some photos of the bones with marks would be interesting. However, we plan to include these pictures in future potential studies. An interesting approach would be specific analyses of the cut marks, seeking to determine whether these were made with flint or metal.

Referrer 2 line 372 Which is Which??? fine line =; gross line=.

Answer: We have completed the information on the caption.

Referrer 2 line 399 This needs ti be supported with additional data / interpretation; and revisited in the discussion.

In this results section there are a lot of suggested hypothesis that need revisiting in the discussion.

Answer: In the results, only the facts and possible explanations are expected to be presented, which are then addressed in the discussion. The problem with zooarchaeological studies in the Bronze Age is that there are very few studies, which is why this research is fascinating. It provides new data that will have to be contrasted with investigations of new sites in the coming years.

Some of these issues are addressed in the discussion.

Referrer 2 line 416-418 These hypotheses need to be better explored. Just because animals / bone elements have no evidence of cut marks, or others, it does. not mean that they were not used for consumption.  The fact that some were fully articulated., demands an overarching interpretation. Also, how do these relate with the human remains found? this is missing from the discussion

Answer: At this point, we only make a renumbering of the aspects identified in the fauna and a declaration of intentions of what we are going to discuss. I wonder why the reviewer wants us to make explanations here when they are aspects that we address later.

In all cases, the reviewer does well to remind us of the association with human remains, which we had forgotten to include here. We have added this data now.

Referrer 2 line 423-424. We should not discard early age death due to other causes than consumption - such as diseases, famine, etc, etc , etc

Answer: It is possible, but if young animals died from disease, they would not have been eaten. The animals found at the site often displayed filleting marks.

Referrer 2 line 439-441 It would be important to provide further supportive arguments and line 446-451: Again: this assumption need to be contextualized, and supportive arguments need to be added.

Answer: We have expanded the discussion to support the opinions shown in this work more robustly with new references.

Referrer 2 line 517-51. Please consider transforming these last paraphrase as major concluding remarks

Answer: Following the reviewer, we have considered this in the final remarks.  

Reviewer 3 Report

Comments and Suggestions for Authors

This is an interesting contribution to the understanding of faunal records in a region in need of quantitative significative assemblages. The paper focuses on the Bronze Age (Bell Beaker and MBA) fauna samples from the Pista de Motos site, in the Madrid region – some minor samples from other chronologies are mentioned. Zooarchaeological and taphonomical studies were made and are presented, also comparing the data at a regional level. This region lacks faunal data for these periods.

Overall, the paper is well prepared in terms of content, although proofreading of several minor formal corrections is needed throughout the paper to be publishable. However, it is scientifically sound and deserves publishing considering the topic addressed, the methodologies applied, and the results obtained. Most figures and tables are informative, with a lot of information being given as Supplementary material. The bibliography is updated. Some minor comments and modifications are suggested below. Double-check the affiliation section according to journal indications. 

The simple summary and abstracts are well prepared. The Introduction section gives a focused overview of data for the Middle Tagus Valley of Inland Iberia.

- Are there no prior studies mentioning the limitation of regional zooarchaeological studies for the Bronze Age? These limitations are seen in other Iberian areas (e.g., almost inexistent in the majority of NW Iberia, very limited in Central and Southern Portugal comparatively to Southern Spain), and relate mostly to the research historically being made or its absence, preservation, and recovery issues. Although this study is focused on inland Iberia (specifically the Middle Tagus Valley), the introduction would benefit from an initial paragraph succinctly describing the broader peninsular data. This would help to reach a wider public and help the reader, especially those less aware of Iberian realities, to better understand the relevance of the analysis presented.

The following section presents the site with some basic information but not the region (as the title suggests).

- Is it possible to have photos and drawings of selected areas of the excavation in the section dealing with the site presentation?

- Mention the linear biometry standards used in the Methods; if they are e.g. Driesch’s, use the adequate English acronyms and nomenclature in corresponding Supplementary Materials tables.

- Regarding age estimates, these approaches have been historically criticized by several experts. Can you acknowledge/discuss a bit the existence of problems with these estimates both in the analytical/register phase and interpretative steps?

- I advise further subdividing the results so it is easier to read and accompany the data presentation.

- The scientific taxonomy needs to be checked throughout the paper: e.g., Sus scropha instead of scrofa, italics being used in sp., Oric instead of Oryc. Lepus europeus instead of europaeus, Suído? etc. The authors use both Equus ferus (article) and Equus caballus (Supplementary Material).

- Uniformize the use of – or / in Table 3, as well as the usage (or not) of capitals in all tables. Table 4: correct the US and EU to SU/SUs.

- Is it possible to make the values in Figures 5 and 6 more clear to the reader?

- Can you further give information on the types of tooth marks registered on lines 319-321? In methodology, the authors mention different typologies, and would be interesting to have some information on their variable frequency.

- Instead of “Prehistorian” Indet. use “Prehistoric” in the tables and text when referring to these samples.

- Line 367 mentions "cylindrically shaped marks". Are these shaft cylinders?

- Explain what the acronyms mean in the tables (e.g., Table 10)

The Discussion and Conclusions present the main interpretations based on the results obtained and compare this data with other regional samples. Finally, the authors also present a paragraph on future directions for analyzing this site’s assemblages, which are very interesting if followed.

- Regarding the comparison with other assemblages, nothing more is added besides lines 457-464 mentioning the sites and the the data in Table 11. Is it possible to further develop this comparison? E.g., why some sites are exceptional regarding the predominant species? Archaeological contexts, location, environmental constraints/characteristics?

- We need to have a map with the sites mentioned in Table 11.

A total of 51 pages are presented as Supplementary material with important information that can be used to understand the assemblage better and for eventual comparison purposes by other colleagues.

- All this needs an important English review throughout. On several occasions, the authors use Spanish words, typos in English words, etc.

Comments on the Quality of English Language

Comments regarding the quality of the English language are addressed in the "Comments and Suggestions for Authors
".

Author Response

Referer 3

Comments and Suggestions for Authors

Referer 3:

This is an interesting contribution to the understanding of faunal records in a region in need of quantitative significative assemblages. The paper focuses on the Bronze Age (Bell Beaker and MBA) fauna samples from the Pista de Motos site, in the Madrid region – some minor samples from other chronologies are mentioned. Zooarchaeological and taphonomical studies were made and are presented, also comparing the data at a regional level. This region lacks faunal data for these periods.

Overall, the paper is well prepared in terms of content, although proofreading of several minor formal corrections is needed throughout the paper to be publishable. However, it is scientifically sound and deserves publishing considering the topic addressed, the methodologies applied, and the results obtained. Most figures and tables are informative, with a lot of information being given as Supplementary material. The bibliography is updated. Some minor comments and modifications are suggested below. Double-check the affiliation section according to journal indications. 

Answer: Many thanks to the reviewer for considering this paper and for the comments that will undoubtedly improve our work.

Referer 3:  The simple summary and abstracts are well prepared. The Introduction section gives a focused overview of data for the Middle Tagus Valley of Inland Iberia.

- Are there no prior studies mentioning the limitation of regional zooarchaeological studies for the Bronze Age? These limitations are seen in other Iberian areas (e.g., almost inexistent in the majority of NW Iberia, very limited in Central and Southern Portugal comparatively to Southern Spain), and relate mostly to the research historically being made or its absence, preservation, and recovery issues. Although this study is focused on inland Iberia (specifically the Middle Tagus Valley), the introduction would benefit from an initial paragraph succinctly describing the broader peninsular data. This would help to reach a wider public and help the reader, especially those less aware of Iberian realities, to better understand the relevance of the analysis presented.

Answer: Following the reviewer's comments, we have added some mentions to the other Iberian regions, thus expanding the contextual framework.

Referer 3:   The following section presents the site with some basic information but not the region (as the title suggests).

Answer: We have added some information about the region.

Referer 3:   Is it possible to have photos and drawings of selected areas of the excavation in the section dealing with the site presentation?

Answer:   We apologise as we currently do not have photos or images of the spatial distribution for the SUs that have been analysed.

Referer 3:  - Mention the linear biometry standards used in the Methods; if they are e.g. Driesch’s, use the adequate English acronyms and nomenclature in corresponding Supplementary Materials tables.

Answer: Following Driesch’s nomenclature, we have changed the acronyms of the SFs.

Referer 3:  Regarding age estimates, these approaches have been historically criticized by several experts. Can you acknowledge/discuss a bit the existence of problems with these estimates both in the analytical/register phase and interpretative steps?

Answer: Following the reviewer, we have added comments on these issues.

- Referer 3:  I advise further subdividing the results so it is easier to read and accompany the data presentation.

Answer: Following the reviewer, we have divided the results into several sections.

- Referer 3: The scientific taxonomy needs to be checked throughout the paper: e.g., Sus scropha instead of scrofa, italics being used in sp., Oric instead of Oryc. Lepus europeus instead of europaeus, Suído? etc. The authors use both Equus ferus (article) and Equus caballus (Supplementary

Material).

 Answer: We have corrected the mistakes.

- Referer 3: Uniformize the use of – or / in Table 3, as well as the usage (or not) of capitals in all tables. Table 4: correct the US and EU to SU/SUs.

Answer: We have corrected the errors

- Referer 3:Is it possible to make the values in Figures 5 and 6 more clear to the reader?

Answer: We have republished images 5 and 6 to make it easier to read.

- Referer 3:Can you further give information on the types of tooth marks registered on lines 319-321? In methodology, the authors mention different typologies, and would be interesting to have some information on their variable frequency.

Answer: The tooth marks are mainly pits and scores. No punctures were documented.

However, we have not documented how many marks appear per bone or other assessments, such as the size of the marks.

We have not analysed the frequency of pits or scores. We have only documented the frequencies of marks that may exist per taxon or anatomical element, which is already shown in Figure 7, and Tables 8 and 9.

- - Referer 3: Instead of “Prehistorian” Indet. use “Prehistoric” in the tables and text when referring to these samples.

Answer: We have corrected the term “prehistorian” to prehistoric.

- Referer 3: Line 367 mentions "cylindrically shaped marks". Are these shaft cylinders?

Answer: We have revised the text and improved the meaning,

- Referer 3:Explain what the acronyms mean in the tables (e.g., Table 10)

Answer: We have completed the acronyms

- Referer 3:The Discussion and Conclusions present the main interpretations based on the results obtained and compare this data with other regional samples. Finally, the authors also present a paragraph on future directions for analyzing this site’s assemblages, which are very interesting if followed.

:Regarding the comparison with other assemblages, nothing more is added besides lines 457-464 mentioning the sites and the the data in Table 11. Is it possible to further develop this comparison? E.g., why some sites are exceptional regarding the predominant species? Archaeological contexts, location, environmental constraints/characteristics?

- We need to have a map with the sites mentioned in Table 11.

Answer: All the sites mentioned are from the same region of the middle Tagus valley, and in principle, the sites are in conditions similar to those of PM.

In any case, we have briefly expanded the discussion in this section and have included a map with the sites.

A total of 51 pages are presented as Supplementary material with important information that can be used to understand the assemblage better and for eventual comparison purposes by other colleagues.

- All this needs an important English review throughout. On several occasions, the authors use Spanish words, typos in English words, etc.

Answer: We have reviewed the supplementary file and corrected the language

Round 2

Reviewer 1 Report

Comments and Suggestions for Authors

It is a pity that a map of the location of the pits and pictures of the skeletons discussed cannot be provided.

2.12.0.0

Reviewer 2 Report

Comments and Suggestions for Authors

Thank you for taking the time to reply to the comments made, which are sufficiently addressed. 

Comments on the Quality of English Language

There is still an issue with the English language, although nothing major.